# Frequency comb generation via synchronous pumped $\chi^{(3)}$ resonator on thin-film lithium niobate

Rebecca Cheng [1] ✉, Mengjie Yu[1,2], Amirhassan Shams-Ansari [1],
Yaowen Hu [1,3], Christian Reimer[4], Mian Zhang[4] & Marko Lončar [1] ✉

Resonator-based optical frequency comb generation is an enabling technology for a myriad of applications ranging from communications to precision spectroscopy. These frequency combs can be generated in nonlinear resonators driven using either continuous-wave (CW) light, which requires alignment of the pump frequency with the cavity resonance, or pulsed light, which also mandates that the pulse repetition rate and cavity free spectral range (FSR) are carefully matched. Advancements in nanophotonics have ignited interest in chip-scale optical frequency combs. However, realizing pulse-driven on-chip Kerr combs remains challenging, as microresonator cavities have limited tuning range in their FSR and resonance frequency. Here, we take steps to overcome this limitation and demonstrate broadband frequency comb generation using a $\chi^{(3)}$ resonator synchronously pumped by a tunable femtosecond pulse generator with on-chip amplitude and phase modulators. Notably, employing pulsed pumping overcomes limitations in Kerr comb generation typically seen in crystalline resonators from stimulated Raman scattering.

With the rapid development of integrated photonics, there has been significant study of on-chip frequency comb generation based on third-order nonlinearity in microresonators[1,2] for applications in communications[3], spectroscopy[4], optical clocks[5], and more[6]. On-chip resonators not only offer much smaller form factor than bulk cavities, but also provide additional opportunities to control the generated output light by leveraging dispersion engineering in addition to phase matching. Kerr comb generation has been demonstrated on a wide range of material platforms, including silica[7], silicon nitride[8], silicon[9,10], aluminum nitride[11], silicon carbide[12], diamond[13], and lithium niobate[14–17]. In most demonstrations, frequency combs are generated using continuous-wave (CW) laser light, requiring high-finesse cavities to achieve the high circulating powers needed for broadband spectrum generation. As a result, many on-chip Kerr combs have repetition rates ranging from 100 s of GHz to THz, outside the range of most high-speed microwave electronics. Additionally, comb generation on crystalline material platforms such as lithium niobate usually suffers from strong Raman gain[18], requiring careful engineering of the free spectral range, pump wavelength, or coupling conditions to mitigate stimulated Raman scattering (SRS)[15,16,19].

Alternatively, these nonlinear resonators can be driven using pulsed laser light via synchronous pumping[20]. Here, the free spectral range (FSR) of the cavity and the repetition rate of the pulse train must also be matched in addition to the pump wavelength to the cavity resonance. Synchronous pumping can lower threshold power for parametric oscillation due to high pulse peak power and has been shown to enable broadband on-chip frequency comb generation of much lower (10 s of GHz) repetition rates[21–23]. Driving with ultrashort pulses could also be an alternative means of mitigating stimulated Raman scattering in microresonators, by simultaneously increasing the SRS threshold when the pulse duration is shorter than the phonon relaxation time[24] and decreasing the four-wave mixing (FWM)

[1]John A. Paulson School of Engineering and Applied Sciences, Harvard University, Cambridge, MA, USA. [2]Ming Hsieh Department of Electrical and Computer Engineering, University of Southern California, Los Angeles, CA, USA. [3]Department of Physics, Harvard University, Cambridge, MA, USA. [4]HyperLight, Cambridge, MA, USA. ✉e-mail: rcheng@g.harvard.edu; loncar@seas.harvard.edu

threshold by seeding the nonlinear process. However, an on-chip pulse-driven comb in which both the pulse generation and comb generation occur on the chip scale remains a large challenge. Unlike bulk oscillators, on-chip resonators have FSRs that are fixed by fabrication with relatively little tunability. Thus, for the realization of a fully chip-scale pulse-driven comb system, it becomes crucial to have an on-chip pulse generator with flexible center frequency and repetition rate. Electro-optic modulation is a well-known method for generating optical pulses from CW laser light with tunable frequency and repetition rate[25–27], and has been implemented with discrete-component modulators to generate tunable pulses for synchronous driving of integrated $\chi^{(2)}$ and $\chi^{(3)}$ resonators[20–23,28].

Thin-film lithium niobate (LN) is an attractive material platform boasting excellent electro-optic and nonlinear optic properties ($r_{33} = 30$ pm/V, $d_{33} = 27$ pm/V, $n_2 = 1.8$e-19 m²/W)[29]. The periodic poling capability of the LN platform also make it appealing for quasi-phase matching of on-chip $\chi^{(2)}$ OPOs or cascaded $\chi^{(2)}$ process for higher effective $\chi^{(3)}$ nonlinearity. Due to these excellent material properties, broadband spectrum generation has been of particular interest on the thin-film LN platform. Previously, ultrashort on-chip pulse generation[30], CW- and synchronous-pumped periodically poled $\chi^{(2)}$ OPO[28,31–34], and Kerr comb generation via $\chi^{(3)}$ OPO[14–17,35] has been demonstrated on TFLN. However, there is currently no work combining on-chip pulse generation with synchronous pumping of resonators using this or any other platform.

In this work, we demonstrate a broad 30-GHz Kerr comb source by leveraging $\chi^{(3)}$ nonlinearity of dispersion-engineered high-$Q$ thin-film LN microresonator that is synchronously pumped using an electro-optic femtosecond pulse source realized via on-chip electro-optic modulation on thin-film LN with pulse compression in optical fiber[30]. The on-chip modulation not only reduces the footprint of our pulsed comb generation scheme, but also gives flexibility of the pulse repetition rate and center wavelength while maintaining small form factor. We find that by employing both synchronous pumping and a microresonator with normal dispersion, we see pump-to-comb efficiency higher than that of traditional CW-pumped Kerr comb generation schemes, which is consistent with previous works focused on studying efficiency in synchronous-pumped and normally dispersive comb generation schemes[22,36]. We also find that pulse-pumped operation significantly mitigates Raman scattering process that has plagued LN[15,18] and other crystalline resonators[19]. Notably, stimulated Raman scattering is well mitigated without any engineering of the free spectral range, pump wavelength, or resonator coupling rates which have been employed in previous works[15,16,19]. The final frequency comb spectrum spans 400 nm over 1400 comb lines (over 2/5 of an octave above the noise floor of our spectrum analyzer, −70 dBm in our case) with a pump-to-comb conversion efficiency of over 10%.

## Results

Our frequency comb is based on $\chi^{(3)}$ four-wave mixing process, which is the dominant process used in traditional microresonator Kerr comb sources[1,2]. Figure 1a illustrates the time- and frequency-domain behavior of comb generation under a typical CW-pumped scheme. In Fig. 1b, we show the comb generation scheme with synchronous pulsed pumping. Pulses are generated from CW light using an electro-optic time lens system through amplitude and phase modulation and dispersion, before being sent to the nonlinear resonator.

Our pulse generator chip is fabricated as described in Ref. 30 on a 600 nm X-cut LN substrate, consisting of one amplitude and one phase modulator with 2-cm long travelling-wave electrodes. The

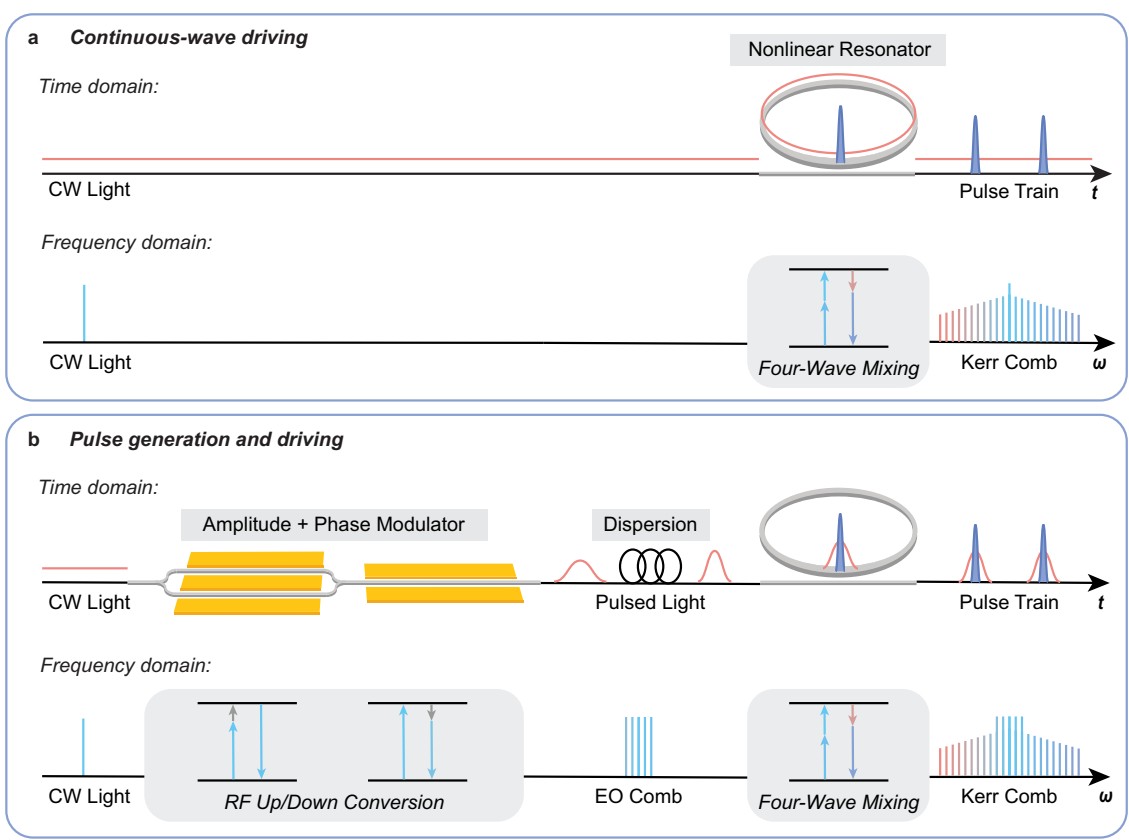

**Fig. 1 | Visualization and concept of pulse and frequency comb generation.** **a** CW scheme in time and frequency domain; CW light is tuned to the resonant frequency of a microresonator and undergoes four-wave mixing process. The generated frequency comb pulse train has a repetition rate equal to the free spectral range of the microresonator; **b** Pulse generation and synchronous pumping scheme in time and frequency domain; a pulse train is formed through amplitude and phase modulation of CW light and compressed in a dispersive medium before resonantly driving the microresonator.

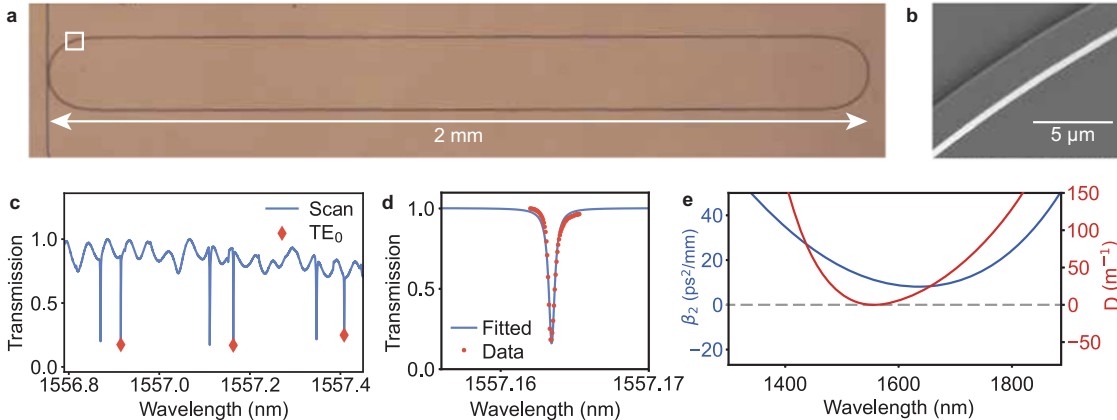

**Fig. 2 | Device images and parameters. a** Optical microscope image of racetrack resonator showing 2 mm footprint size. **b** Scanning electron microscope (SEM) image of waveguide sidewalls in the bending region of the racetrack resonator. **c** Wide resonance scan (blue) showing FSR of 30.14 GHz for the fundamental TE mode (indicated in red) around 1557 nm. **d** Resonance linewidth fitting of fundamental TE mode, showing loaded optical Q-factor of 3.2 million and linewidth of ~60 MHz. **e** Dispersion profiles of fabricated geometry with pumping at the resonance frequency.

**Table 1 | Performance comparison with other LN Kerr combs**

| Reference | Repetition rate (GHz) | Center wavelength (nm) | On-chip power (mW) | Pump scheme | Bandwidth (octave) | Dispersion | Mitigated SRS |
|---|---|---|---|---|---|---|---|
| This work | 30 | 1557 | 12.5 | Pulsed | 0.4 | Normal | Yes |
| Ref 17 | 250 | 1550 | 300 | CW | 0.65 | Anomalous | No |
| Ref 15 | 335 | 1550 | 240 | CW | 0.8 | Anomalous | Yes |
| Ref 16 | 200 | 2000 | 90 | CW | 0.3 | Anomalous | Yes |
| Ref 14 | 200 | 1550 | 33 | CW | 0.25 | Anomalous | Yes |
| Ref 35 | 200 | 1550 | 600 | CW | 1.15 | Anomalous | Yes |

fiber-to-fiber insertion loss of this chip is 9 dB (3 dB/facet for each facet and 3 dB loss from device operation). The pulse generator chip features a recycled-PM design to lower the $V_\pi$ of the phase modulator. As such, the optimal operating condition is achieved when the optical signal is in phase with the microwave drive and the phase modulator exhibits resonant behavior, hitting the lowest $V_\pi$ every 2.8 GHz. One such resonant operating frequency occurs at 30.135 GHz. When operated at this RF frequency, the generated pulses reach maximum compression after passing through 59-m of single-mode fiber and achieve a pulse duration of 526 fs[30].

The resonator (Fig. 2a, b) is fabricated on Z-cut LN in order to avoid TE/TM mode crossing in the racetrack bends caused by material birefringence. The racetrack is designed so that the FSR of its fundamental TE mode is within the resonant operating range of the EO pulse generator, measured to be 30.14 GHz (Fig. 2c). After the optical layers are defined, the device then undergoes a 2-h annealing process at atmospheric pressure with $O_2$ ambient, intended to decrease the thin film material absorption limit and increase the optical quality factors of our resonator device[37]. Accordingly, the waveguides are designed to be wide and multimode (2 μm) so that the optical quality factor is not limited by fabrication-induced sidewall scattering. The fundamental TE mode of the resonator is near-critically coupled and has a loaded optical linewidth of 60 MHz and optical Q-factors of 3.2 million (Fig. 2d). We note a near 5× improvement in the Q of this device compared to our previous 250 GHz Kerr comb devices[17], made possible by this annealing process. The calculated dispersion profiles for our waveguide geometry are given in Fig. 2e (see Figure S1 for the dispersion profiles before and after annealing). The insertion loss of this chip is 11 dB (5.5 dB/facet for each facet and negligible on-chip loss).

For the measurement (Fig. 3a, Fig. S2), light is coupled between the two LN chips using lensed fiber and single-mode fiber (SMF-28). The total length of SMF between the two devices is carefully calibrated

to reach maximum pulse compression. An erbium-doped fiber amplifier (EDFA) is used to amplify the pulse between the two chips. The center frequency of the pulse source is tuned to match the resonance frequency, and its repetition rate is fine-tuned to precisely match the FSR of the Kerr ring with 1 MHz accuracy (see Fig. S3). The broadening behavior under varying optical power is shown in Fig. S4.

For generation of the frequency comb spectrum, the EDFA gain is set to 12 dB. The gain is largely used to compensate for the chip-chip coupling loss (3 dB at the output of EO pulse generator, and 5.5 dB at the input of the resonator) and supplies an additional 3.5 dB of gain to the system. The generated comb spectrum is given in Fig. 3b. The asymmetric profile of the frequency comb reflects the dispersion operator curve given in Fig. 2e, which biases the comb in the red direction with respect to the pump. The final frequency comb is achieved using ~12.5 mW of average on-chip power, corresponding to 0.8 W of peak power and 0.45 pJ pulse energy, in the bus waveguide, and spans 2/5 of an octave over 1400 comb lines. The conversion efficiency of our frequency comb, which we define as the total power of the frequency comb divided by the average power in the bus waveguide, is measured to be ~10%. The high efficiency is enabled by both synchronous driving as well as normally dispersive operation[22]. To date, this is the highest reported pump-to-comb conversion efficiency of Kerr comb on thin-film LN. Key features of the comb, including bandwidth, center wavelength, and repetition rate, are compared against other LN Kerr combs in Table 1.

Importantly, synchronous driving overcomes strong stimulated Raman scattering (SRS) in the material, another limitation in LN-based frequency combs and $\chi^{(3)}$ OPOs based on crystalline materials[18]. SRS has been observed in both X and Z crystal cuts of thin-film lithium niobate and extensively studied along both crystal orientations in X-cut LN[18], which finds that light generated through SRS in LN scatters in the backward direction to the optical pump direction. In our pulsed

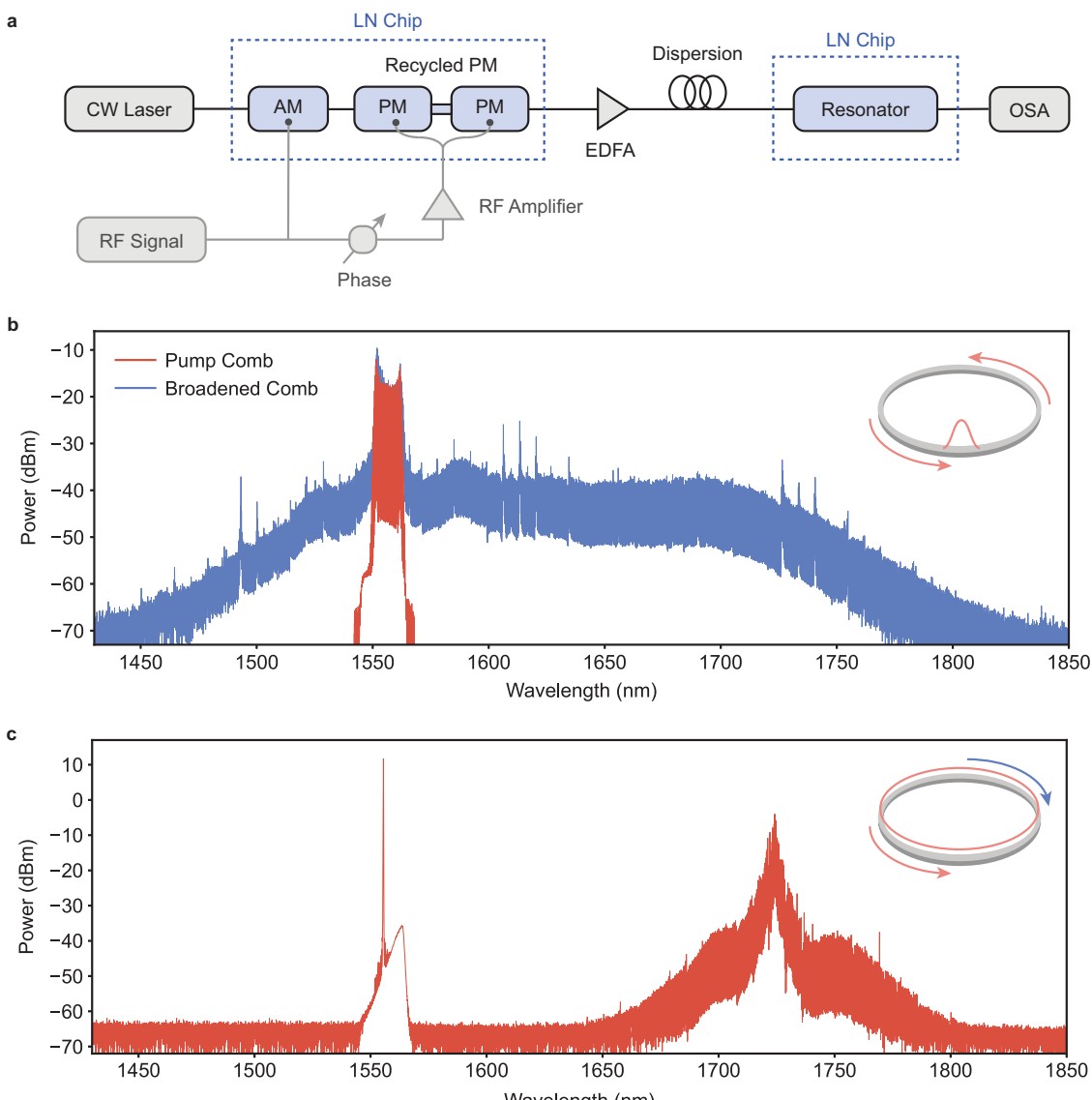

**Fig. 3 | Frequency comb generation in pulse-driven vs. CW-driven case.**
**a** Measurement scheme for on-chip pulse-pumped frequency comb. CW light is coupled onto the first LN chip for pulse generation. The outgoing pulse train is amplified using an EDFA and compressed in single mode fiber. The pulsed light is then coupled to the second LN chip for spectral broadening; **b** Pulse-pumped spectrum, generated with 12.5 mW average on-chip power. The spectrum spans 2/5 octave. Red trace: electro-optic frequency comb produced by the pulse generator chip, measured before coupling into the resonator chip. Blue trace: frequency comb spectrum produced via pulse-pumping, measured at the resonator chip output. The power level of the red trace is adjusted to account for chip-chip coupling loss, so that the relative power of the pump comb and broadened comb match appropriately. Inset: visualization of light circulation in microresonator for synchronous driving. SRS is avoided and FWM co-propagates with the pump; **c** CW-pumped spectrum, generated with 200 mW on-chip power, monitored in the backward direction. A strong Raman peak is observed at 1720 nm. Inset: visualization of light circulation. SRS threshold is lower than that for FWM, and so both SRS and subsequent comb lines generated through FWM counter-propagate with the pump.

scheme, we excite the resonator synchronously with an EO-generated comb, thus seeding and lowering the power threshold of the four-wave mixing process. This favors the Kerr comb generation in the forward direction over strong Raman scattering followed by Kerr comb generation in the backward direction. To confirm this experimentally, the LN resonator is also excited using a high-power CW source. Figure 3c shows the generated comb spectrum at 200 mW of on-chip power in the CW driving case, monitored in the backward direction. We see a strong peak in the comb spectrum at ~1725 nm, corresponding to a Raman frequency shift of 625 cm⁻¹ and the resonance of the E(TO)₉ phonon branch[38–42]. Moreover, the microcomb that forms around the peak at 1725 nm is significantly stronger than the comb lines around the pump, similar to previously observed Raman frequency combs in

LN and further supporting the observance of Raman scattering under this generation scheme. To our knowledge, this is the first demonstration of Raman suppression through pulsed pumping.

## Discussion

In conclusion, we show a synchronously driven LN resonator and resonant Kerr broadening of an electro-optic optic pulse source on thin-film lithium niobate, generating a 30-GHz comb spanning 400 nm in bandwidth. Additionally, the seeded four-wave mixing process of the pulsed-pumping scheme greatly reduces the threshold power for comb generation and bypasses the Raman scattering process commonly observed in thin-film LN comb generation. The mitigation of stimulated Raman scattering allows for the generation of broadband

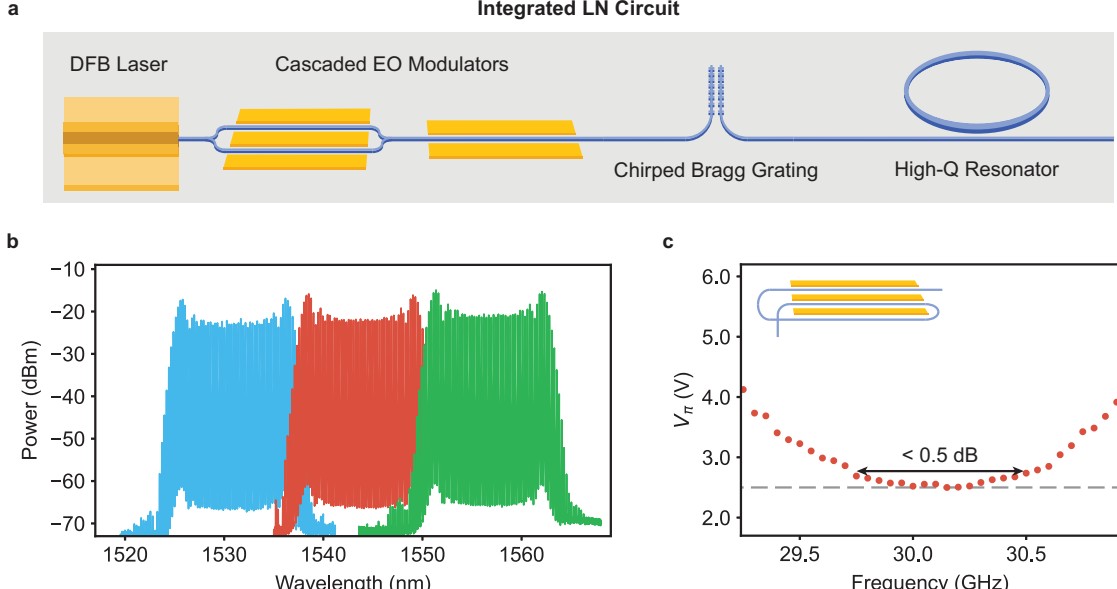

**Fig. 4 | Outlook of synchronously driven resonators on thin-film LN.**
**a** Schematic for integrated on-chip pulse-driven comb generation, which includes integrated on-chip DFB laser source, amplitude and phase modulators for pulse generation, on-chip dispersion management, and dispersion engineered $\chi^{(2)}$ or $\chi^{(3)}$ resonators. In this schematic, the proposed dispersion compensation is implemented using a coupled chirped Bragg grating system[30,50]. For future integrated schemes, fabrication developments can be made on modulator side (Z-cut modulators) or on the resonator side (dispersion engineering with respect to material birefringence) for compatibility of integration; **b** Pulse generation at different

center wavelengths. The pulse source has flexible wavelength tunability and can be precisely matched to resonator frequency without large need for thermal tuning; **c** $V_\pi$ of the recycled phase modulator (layout shown in inset) utilized in the pulse generator design. The waveguide wraps around before passing through modulation again, allowing the phase modulator to be driven with a single GSG electrode and microwave source but imparting a periodic resonance condition on the $V_\pi$. The $V_\pi$ is measured around the resonance condition 30.135 GHz (2.5 V). Within the range that we can fabricate reproducibly, indicated by black arrow, there is <0.5 dB change in the $V_\pi$ and minor effect on the pulse generation efficiency.

LN optical frequency combs at lower repetition rates. For applications requiring high conversion efficiencies, pump-to-comb conversion efficiency can be improved further through operating the resonator in the over coupled regime[22] or through the introduction of an auxiliary resonator[43]. Careful study of the interplay between stimulated Raman scattering and soliton formation under pulsed pumping could be explored further for even broader bandwidth combs in the normally dispersive regime[44,45].

We believe that the thin-film LN platform is a promising host for on-chip nonlinear optics beyond what has been demonstrated in this work. The poling capability of LN makes it an excellent candidate for $\chi^{(2)}$ pulsed optical parametric oscillators, and $\chi^{(3)}$ Kerr combs operating in the anomalous dispersion regime for octave spanning comb generation are also a candidate for synchronous driving. While our current measurement was done across two separate chips, we look forward to future compatibility of this scheme for full integration (Fig. 4a). This would not only greatly reduce the footprint of the comb generator, but also allow us to avoid on- and off-chip coupling losses between the two chips. Combined with higher pump-power lasers, this loss reduction would also eliminate the need for an intra-system amplifier. Platform compatibility between the two chips should be possible with careful design and fabrication development. For example, the electro-optic components can be made on Z-cut LN with development of high efficiency Z-cut LN modulators[46]. On the other hand, the resonator chip could be fabricated on X-cut LN through careful engineering of the racetrack bends to avoid mode crossing caused by material birefringence[47].

While fabrication across two chips allows flexibility to sweep a range of device sizes to precisely match the optimal repetition rate of the pulse generator, the on-chip pulsed resonator scheme should be robust to fabrication should the two components be combined. The center wavelength of the pulse source is flexible (Fig. 4b), as the

recycled phase modulator is not optically resonant and the quadrature point of the amplitude modulator could be tuned with DC bias (electro-optics) or heaters (thermo-optics). Additionally, we conservatively estimate that we can target the resonance FSR within 300 MHz of the intended design given fabrication variation and error. We find that within this RF frequency range, there is negligible ( ~1%) change in the $V_\pi$ of the recycled phase modulator (Fig. 4c).

An outstanding bottleneck to a fully integrated scheme not demonstrated in this work is the need for on-chip dispersion compensation to compress the pulses generated by the modulator system. One approach to on-chip dispersion is through the use of a long waveguide with large engineered dispersion. However, because of the waveguide geometries usually needed to achieve high dispersion, this approach is limited largely by the propagation loss in these waveguides, currently much higher than the state-of-the-art TFLN propagation loss[30]. Waveguide gratings are a more commonly used method for dispersion compensation on integrated photonics platforms and can achieve large dispersion factors over a small propagation distance[48,49]; they have been implemented on the TFLN platform with low insertion loss and small footprint[30]. Because these gratings operate in the reflection mode, most implementations of on-chip dispersion gratings require optical circulators or beam splitters. Recently, circulator-free dispersion schemes based on coupled waveguide gratings[50] (used to illustrate a potential approach to on-chip dispersion in Fig. 4a) and mode de-multiplexers[51] have been proposed and demonstrated.

Recently, in addition to pulse generation, compression, and nonlinear broadening, efforts in integrating high power lasers have been demonstrated on thin-film lithium niobate[52]. We look forward to a fully integrated scheme that combines not only the pulse generator and resonator, but also on-chip pulse compression and laser integration. We envision a scheme that combines all these components

together, which could be possible with further fabrication development, improvements in loss, and larger scale processing of lithium niobate components.

## Methods

### Device fabrication

The electro-optic pulse generator device is fabricated on thin-film X-cut lithium niobate on insulator wafer (NanoLN) with 600 nm film thickness and and buried oxide thickness of 2 µm. The resonator device is fabricated on thin-film Z-cut lithium niobate on insulator wafer (also NanoLN) with 600 nm film thickness and 2 µm buried oxide thickness. The optical layer for devices are defined using electron-beam lithography (EBL) with hydrogen silsesquioxane resist, which is then partially etched by Ar + -based reactive ion etching. The X-cut device is partially etched to 300 nm with a remaining slab of 300 nm. The entire device is cladded with silicon dioxide via plasma-enhanced chemical vapor deposition (PECVD). The microwave electrodes (1.6 µm Au) are defined using electron-beam lithography and photolithography and metallized using electron beam evaporation. The facets of the X-cut LN chip are etched with deep reactive ion etching. The Z-cut device is partially etched by 485 nm with a remaining slab of 115 nm. The chip undergoes 2-hours of high temperature furnace annealing in $O_2$ ambient. The facets of the Z-cut LN chip are cleaved to ensure a smooth facet to minimize coupling loss.

### Measurement of the resonator spectrum in pulse- and CW-driven cases

For pulsed frequency comb characterization, continuous-wave light from a tunable telecom laser (Santec TSL-570) is coupled into the X-cut modulator chip used tapered lensed fiber. A three-paddle fiber polarization controller is used to control the polarization of input light. The modulators on the LN chip are driven using dual ground-signal-ground (GSG) probes (GGB Industries) with a signal generator (Keysight) split into two arms. In the amplitude modulator arm, the signal is sent through a pre-amplifier and variable attenuator (RF Lambda). For the phase modulator, the signal goes through a phase shifter and high-power amplifier (RF Lambda). The output light (collected with lensed fiber) is sent into an erbium-doped fiber amplifier (EDFA, ThorLabs) and another fiber polarization controller before being coupled onto the Z-cut resonator chip. The total fiber length between the setups, including the fiber length in the polarization paddles and EDFA, is 59 m. The output light is sent into an optical spectrum analyzer (Yokogawa) for analysis.

For continuous-wave driving, the tunable laser is sent into a high-power EDFA (Ammonics) and coupled to the Z-cut resonator chip using lensed fiber. A polarization controller is used to control the input light and an optical circulator is used to monitor both the forward and backward propagated light from the chip. The backward light (circulator port 3) is sent into an optical spectrum analyzer for analysis.

Please see the Supplementary Information and Fig. S2 for a full measurement schematic for spectrum characterization.

## Data availability

All data supporting the findings of the study are found in the Article and Supplementary Information. The data required to generate the figures in this work are available at https://doi.org/10.6084/m9.figshare.25674741[53].

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

## Acknowledgements
This work is supported by the Defense Advanced Research Projects Agency HR0011-20-C-0137 (R.C., M.Y., A.S.-A., Y.H., C.R. M.Z., and M.L.), ONR N00014-18-C-1043 (R.C. and M.Y.), ONR N00014-22-C-1041 (R.C. and M.Y), and AFOSR FA9550-19-1-0376 (A.S.-A.). This work was performed in part at the Harvard University Center for Nanoscale Systems (CNS); a member of the National Nanotechnology Coordinated Infrastructure Network (NNCI), which is supported by the National Science Foundation under NSF Award ECCS-2025158. The views, opinions and/ or findings expressed are those of the author and should not be interpreted as representing the official views or policies of the Department of Defense or the U.S. Government.

## Author contributions
M.Y. conceived the idea. R.C. designed, fabricated, and characterized the resonator chip. M.Y. designed and characterized the pulse generator chip. C.R. and M.Z. fabricated the pulse generator chip. R.C. and M.Y. carried out the measurement. A.S.-A. and Y.H. helped with the project. R.C. analyzed the data and wrote the manuscript with contribution from all authors. M.L. supervised the project.

## Competing interests
C.R., M.Z. and M.L. are involved in developing lithium niobate technologies at HyperLight Corporation. The remaining authors declare no competing interests.
