## [Peer Review File · Nature Communications]

REVIEWER COMMENTS

Reviewer #1 (Remarks to the Author):

The manuscript describes the experimental realization of a frequency comb source with a 30 GHz repetition rate with a -10 dB bandwidth of about 30 THz at the central frequency of 187 THz, using χ^3 interaction in a z-cut LiNbO₃ racetrack resonator and synchronous pumping. Although the title and the introduction might give an impression that this is a monolithic "on-chip" device, which would be a real breakthrough, in reality, the pump source and the FWM ring are connected through a fiber amplifier and a dispersive delay line for the EO comb compression. The EO pump source has already been published in Ref. 20.

It is not evident that the functions of the sub-ps pump source and the FWM comb generator could be easily combined on an LN chip of the same cut. In many aspects, the work is similar to that in Ref.10, except for the fact that in the current work, a positive GVD regime was used. Getting away from the soliton regime has some potential advantages regarding power (and efficiency) scaling, although that has not been demonstrated in this work. Another advantage of the pulsed pumping for LN and other crystalline host resonators is related to the increase of the stimulated Raman scattering threshold, which is demonstrated in the manuscript. The manuscript is clearly written, although the work could be better documented (see comments below). Overall, the progress reported in the work lacks the breakthrough character.

Specific comments:

1. The term "oscillator" typically implies the existence of an oscillation threshold. For instance, the first FWM OPO reported in Opt. Lett. 27, 1675 (2002) or more recent Phys. Rev. Lett. 129, 163601 (2022) makes this point very clear in the FWM context. In the current work, the threshold is not specified.
2. It is not clear how the bandwidth of the 2/5 octave was determined. The commonly used reference levels of -3dB or -10dB would give substantially narrower bandwidth.
3. Fig. 2(c) displays a spectrum over several resonances around 1557 nm. The red coupling dips are specified as TE₀; the blue dips are called "scan". Unclear what that means and how that spectrum was measured.

Reviewer #2 (Remarks to the Author):

The article "On-chip synchronous pumped χ^3 optical parametric oscillator on thin-film lithium niobate" reports on a chip-integrated set of electro-optic modulators providing an electro-optic frequency comb. This comb is amplified in an off-chip amplifier and temporally compressed in 59 meters of optical fiber. The resulting pulses are coupled to a second chip with a microcavity that is synchronously driven by the previously generated pulses, resulting in broadband comb generation.

All subcomponents of the system, including the on-chip modulators and microcavity, have been demonstrated previously by the same group, and driving a microcavity with an electro-optic pulse train is well established. However, demonstrating two key components, modulation stage and microcavity on the same platform, and that they work together overcoming undesirable dynamics originating from the

Raman effect is an important result.

Before I would recommend the manuscript for publication in Nature Communications the following points should be addressed:

- In the introduction a reference to original work on electro-optic pulse generation (<https://doi.org/10.1109/3.135>) and previous work on a synchronously-pumped Lithium-Niobate microcavity OPO is missing (<https://doi.org/10.1364/OL.43.005745>). The authors mention the optical conversion efficiency in their system to be around 10% and state that this would overcome efficiency limitations in traditional CW-pumped Kerr combs. This should be placed into context with previous work on efficient systems with 30% efficiency in the normal (<https://doi.org/10.1002/lpor.201600276>) and 50% in the anomalous dispersion regime(<https://arxiv.org/abs/2202.09410>).
- The authors state that their work is the first one combining on-chip pulse generation with on-chip optical parametric oscillations. I find this statement misleading, as the pulse generator is not fully on-chip but includes off-chip compression and amplification. It would perhaps be more accurate to say that the current setup includes on-chip electro-optic modulators and an on-chip microcavity that are implemented in the same thin-film platform, and that the authors have ideas how to integrate all other components in the future. It is further stated that the on-chip pulse source enabled tuning of pulse repetition rate and center wavelength. This is also the case for conventional electro-optic modulators and not an advantage of the present system.
- The authors describe how the pulse driving overcomes certain limitations related to the Raman effect. In my view, this is an important insight and distinguishes the present work from previous work. A more detailed discussion would be in place: How many pump lines are needed to favor FWM over SRS? How much SRS is tolerable, or, as demonstrated previously even beneficial (<https://doi.org/10.1364/OL.413585>)? Why do the authors expect SRS to be in the backward direction?
- It is described that the repetition rate is fine-tuned to match the microcavity's FSR. How delicate is this tuning and what are the tolerances, also regarding thermal drifts etc.? Perhaps the resonator transmission and spectra for different repetition rates/center wavelength detuning could be shown?
- Is there any enhancement of the effective χ^3 nonlinearity through cascaded χ^2 effects?

Response to the Reviewers

We would like to thank the reviewers for their time in reading our manuscript and providing valuable feedback. We have made changes based on their feedback that we believe makes our article stronger. These changes include updated measurements on our system. We note minor discrepancies in the dispersion of our resonator in the updated measurements (slightly different spectrum profile and a 1 MHz change in the free spectral range) that we attribute to minor aging of the sample over time. We believe that the scientific insights of these measurements are still valid despite this change. This point is reflected in our manuscript as such.

We address the reviewers point by point below.

Reviewer #1 (Remarks to the Author):

The manuscript describes the experimental realization of a frequency comb source with a 30 GHz repetition rate with a -10 dB bandwidth of about 30 THz at the central frequency of 187 THz, using χ^3 interaction in a z-cut LiNbO₃ racetrack resonator and synchronous pumping. Although the title and the introduction might give an impression that this is a monolithic "on-chip" device, which would be a real breakthrough, in reality, the pump source and the FWM ring are connected through a fiber amplifier and a dispersive delay line for the EO comb compression. The EO pump source has already been published in Ref. 20.

We thank the reviewer for the comment and for pointing out the potentially misleading title of our manuscript. To avoid confusion, we have changed the manuscript title to: *Frequency comb generation via synchronous pumped resonator on thin-film lithium niobate*, removing "on-chip" and focusing on the individual components in the body of the manuscript. Additionally, based on feedback from both reviewers, we shift the framing of the manuscript away from optical parametric oscillation and towards the mitigation of stimulated Raman scattering via pulsed pumping for frequency comb generation.

It is not evident that the functions of the sub-ps pump source and the FWM comb generator could be easily combined on an LN chip of the same cut. In many aspects, the work is similar to that in Ref.10, except for the fact that in the current work, a positive GVD regime was used. Getting away from the soliton regime has some potential advantages regarding power (and efficiency) scaling, although that has not been demonstrated in this work.

We would like to push back on the claim that our work is no different than that of Ref. 10 besides the GVD regime used, for three main reasons. The first is that while the other manuscript does also electro-optic modulation generated pulses to pump a resonator, their work uses off-chip and bulk modulators. We believe that our demonstration of both on-chip pulse generation interfaced with on-chip resonator is a crucial steppingstone for future fully integrated schematics. The second, which relates to the first point, is the material platform (lithium niobate as opposed to silicon nitride). LN is an excellent host platform for nonlinear optics, not only due to the qualities of the resonators demonstrated in our manuscript and the excellent electro-optic response, but also it is compatible with frequency doubling schemes that are often used for self-referenced combs. Finally, relatedly and perhaps somewhat philosophically, there are many micro-resonator based CW pumped frequency combs reported in published in variety of

platforms (Si, SiN, SiO₂, diamond, LN, AlN...) and they all rely on more or less same physics. But each platform offers something different and new.

Another advantage of the pulsed pumping for LN and other crystalline host resonators is related to the increase of the stimulated Raman scattering threshold, which is demonstrated in the manuscript. The manuscript is clearly written, although the work could be better documented (see comments below). Overall, the progress reported in the work lacks the breakthrough character.

We thank the reviewer for pointing out the value of using pulsed-pumping schemes for overcoming stimulated Raman scattering which has been a bottleneck for crystalline resonators. We hope that with the changes made to the wording of the manuscript, additional measurements, and focus on overcoming Raman limitations, the reviewer agrees that our work is suitable for publication.

Specific comments:

1. The term "oscillator" typically implies the existence of an oscillation threshold. For instance, the first FWM OPO reported in *Opt. Lett.* 27, 1675 (2002) or more recent *Phys. Rev. Lett.* 129, 163601 (2022) makes this point very clear in the FWM context. In the current work, the threshold is not specified.

We thank the reviewer for the comment, which has sparked some discussion on the wording of our manuscript. While performing optical-power dependent measurements to respond to the reviewer's point, we discovered that our resonator does not exhibit typical optical parametric oscillation behavior and instead broadens with increasing optical power due to self-phase modulation instead of degenerate four-wave mixing. To address this point, we have reframed the manuscript slightly to reflect pulse-pumped resonators for frequency comb generation rather than optical parametric oscillation. We choose to include optical power-dependence measurements in the Supplementary Material.

For our measurement, the optical power is tuned by adjusting the current on the EDFA, which has no effect on the pulse compression between the setups. The on-chip power is calculated using the output power after the resonator chip combined with measured per facet coupling losses. We explore a range of average on-chip powers from 1.0 to 5.0 mW. We note that under continuous wave (CW) operation, the stimulated Raman scattering dominates over other nonlinear processes, so we have only included the optical power measurement under pulsed operation. The manuscript text mentions the measurement and redirects the reader to the Supplementary Material as modified below:

“An erbium-doped fiber amplifier (EDFA) is used to amplify the pulse between the two chips. The center frequency of the pulse source is tuned to match the resonance frequency, and its repetition rate is fine-tuned to precisely match the FSR of the Kerr ring (see Methods). **As optical power increases, the spectrum broadens around the pump EO comb. The optical spectrum under varying optical power is given in Methods.**” Additionally, the figure for the supplementary material is included below as well. The text detailing the measurement can be found in the methods section titled “Optical power.”

Response Figure 1. Generated optical spectra for average on-chip powers ranging from 1.0 to 5.0 mW. The power is adjusted by tuning the current on the EDFA. As expected, we see broader comb spectrum for increasing optical power.

2. It is not clear how the bandwidth of the 2/5 octave was determined. The commonly used reference levels of -3dB or -10dB would give substantially narrower bandwidth.

The 2/5 octave bandwidth is determined by using the span of the comb lines above the noise floor of our optical spectrum analyzer (-70 dBm in our case). We note that previously published works on microcombs and synchronously pumped resonators appear to calculate the bandwidth using a similar metric^{1,2}. However, to make this clearer to the reader, we adjusted our text to clarify how the 2/5 of an octave was calculated. We have also removed claims of the comb bandwidth from our abstract to avoid misleading the reader.

The last sentence of the introduction is modified to read: “The final frequency comb spectrum spans 400 nm over 1400 comb lines (over 2/5 of an octave above the noise floor of our spectrum analyzer, -70 dBm in our case) with a pump-to-comb conversion efficiency of over 10%.”

3. Fig. 2(c) displays a spectrum over several resonances around 1557 nm. The red coupling dips are specified as TE₀; the blue dips are called "scan". Unclear what that means and how that spectrum was measured.

We apologize for the confusion. To clarify for the reviewer, the resonance “scan”, which is plotted in blue, contains the transmission spectrum, including the fundamental TE₀ mode as well as a higher order TE₁ mode. The red is meant to highlight the fact that the TE₀ modes are the ones of interest, although the wide width of our resonator does support higher order modes as well. We now see the potential misleading nature of the figure and have modified the means of plotting accordingly. Please see the modified figure and caption below:

Fig. 2 Device images and parameters. **a**, optical microscope image of racetrack resonator showing 2 mm footprint size. **b**, scanning electron microscope (SEM) image of waveguide sidewalls in the bending region of the racetrack resonator. **c**, wide resonance scan (blue) showing FSR of 30.14 GHz for the fundamental TE mode (indicated in red) around 1557 nm. **d**, resonance linewidth fitting of fundamental TE mode, showing loaded optical Q-factor of 3.2 million and linewidth of ~60 MHz. **e**, dispersion profiles of fabricated geometry with pumping at the resonance frequency.

Reviewer #2 (Remarks to the Author):

The article "On-chip synchronous pumped χ^3 optical parametric oscillator on thin-film lithium niobate" reports on a chip-integrated set of electro-optic modulators providing an electro-optic frequency comb. This comb is amplified in an off-chip amplifier and temporally compressed in 59 meters of optical fiber. The resulting pulses are coupled to a second chip with a microcavity that is synchronously driven by the previously generated pulses, resulting in broadband comb generation.

All subcomponents of the system, including the on-chip modulators and microcavity, have been demonstrated previously by the same group, and driving a microcavity with an electro-optic pulse train is well established. However, demonstrating two key components, modulation stage and microcavity on the same platform, and that they work together overcoming undesirable dynamics originating from the Raman effect is an important result.

We thank the reviewer for this comment, and hope that they can be convinced that our work can be recommended for publication in Nature Communications with the modification we have made to the manuscript. Our detailed response is below:

Before I would recommend the manuscript for publication in Nature Communications the following points should be addressed:

- In the introduction a reference to original work on electro-optic pulse generation (<https://doi.org/10.1109/3.135>) and previous work on a synchronously-pumped Lithium-Niobate microcavity OPO is missing (<https://doi.org/10.1364/OL.43.005745>). The authors mention the optical conversion efficiency in their system to be around 10% and state that this would overcome efficiency limitations in traditional CW-pumped Kerr combs. This should be placed into context with previous work on efficient systems with 30% efficiency in the normal (<https://doi.org/10.1002/lpor.201600276>) and 50% in the anomalous dispersion regime (<https://arxiv.org/abs/2202.09410>).

We thank the reviewer for drawing our attention to excellent work that is currently missing in our reference list. We have modified parts of the introduction text to reflect this, including the reference on electro-optic pulse generation with the other works cited as well as the previous work on $\chi^{(3)}$ LN OPO with the other LN OPO works, as well as placing the statements on efficiency against the works that the reviewer has cited. The changes in the text are included below:

To allow for the inclusion of the synchronously-pumped LN OPO with our other references, we make small changes: "Electro-optic modulation is a well-known method for generating optical pulses [...] and has been implemented with discrete-component modulators to generate tunable pulses for synchronous driving of integrated $\chi^{(2)}$ and $\chi^{(3)}$ resonators." And, "Previously, ultrashort on-chip pulse generation via CW- and synchronous-pumped periodically poled $\chi^{(2)}$ OPO [...] has been demonstrated on the thin-film LN platform."

To place the statements of efficiency in better context with works meant to study and optimize conversion efficiency, we write: "We find that by employing both synchronous pumping and a microresonator with normal dispersion, we see pump-to-comb efficiency higher

than that of traditional CW-pumped Kerr comb generation schemes, which is consistent with previous works focused on studying efficiency in synchronous-pumped and normally dispersive comb generation schemes.”

We believe that the work that the reviewer cites which achieves 50% conversion efficiency in the anomalously dispersive regime is not a fair comparison and are not included in our statements about “traditional CW-pumped Kerr comb schemes.” The work uses an auxiliary cavity to tune the comb coupling rate from the bus waveguide, not unlike other works using coupled ring systems, but decidedly different from the single ring schemes that are cited throughout our manuscript. In order to include this reference in the proper context, we have chosen to add this reference (with additional statements) to the conclusion and outlook: “For applications requiring high conversion efficiencies, pump-to-comb conversion efficiency can be improved further through operating the resonator in the overcoupled regime or through the introduction of an auxiliary resonator.” We hope the reviewer is amenable to this decision.

- The authors state that their work is the first one combining on-chip pulse generation with on-chip optical parametric oscillations. I find this statement misleading, as the pulse generator is not fully on-chip but includes off-chip compression and amplification. It would perhaps be more accurate to say that the current setup includes on-chip electro-optic modulators and an on-chip microcavity that are implemented in the same thin-film platform, and that the authors have ideas how to integrate all other components in the future. It is further stated that the on-chip pulse source enabled tuning of pulse repetition rate and center wavelength. This is also the case for conventional electro-optic modulators and not an advantage of the present system.

To avoid misleading the reader on the scope of our work, we have modified the text to avoid the claim that our work fully combines on-chip pulse generation with on-chip Kerr frequency comb generation. Please see also our note to Reviewer #1, where we have modified the manuscript title to further avoid confusion. Additionally, our abstract now reads: “Here, we take steps to overcome this limitation and demonstrate broadband frequency comb generation using a $\chi^{(3)}$ pulse-driven resonator by using a tunable on-chip femtosecond pulse generator to synchronously pump the ring.” to avoid the claim that an on-chip pulse-driven resonator is fully demonstrated.

We agree with the reviewer that conventional EO modulators provide the same benefit of pulse repetition rate and center wavelength tunability. In our writing of the manuscript, we did not intend to suggest the opposite. We have modified the text to reflect the advantage of the on-chip pulse source in repetition rate and wavelength tunability *over other on-chip pulse generation solutions*, rather than to imply that there is novelty in tunability by moving on chip over conventional bulk modulators: “Thus, for the realization of a fully chip-scale pulse-driven comb system, it becomes crucial to have an on-chip pulse generator with flexible center frequency and repetition rate.” And then later in the manuscript text: “The on-chip pulse source not only reduces the footprint of our pulsed comb generation scheme, but also gives flexibility of the pulse repetition rate and center wavelength while maintaining small form factor.”

- The authors describe how the pulse driving overcomes certain limitations related to the Raman effect. In my view, this is an important insight and distinguishes the present work from previous work. A more detailed discussion would be in place: How many pump lines are needed to favor FWM over SRS? How much SRS is tolerable, or, as demonstrated previously even beneficial (<https://doi.org/10.1364/OL.413585>)?

We thank the reviewer for the comment and insights. Indeed, a careful study of comb-line dependence on the interplay of SRS and FWM would be interesting, and we have thought carefully about how a measurement can be set up to address the reviewer's point.

We would like to point out difficulty in fully decoupling the number of pump lines from our measurement scheme. The number of pump lines can be tuned by decreasing the driving strength of the phase modulator in the pulse generation chip, but this also effects the quadratic phase imparted on the pulses carved by the amplitude modulator, changing the dispersion required as well as the minimum pulse duration achievable³. Notably, as the phase modulation index and number of comb lines go down, so does the peak power, so cases with fewer comb lines may not reach the Raman oscillation threshold in our measurement.

However, to partially answer the reviewer's question, we include a measurement that shows intermediate steps in the reduction of comb lines for pumping the resonator. The electro-optic comb generated in our manuscript spans about 40 comb lines, excluding the tails (see insets in Response Figure 2). We include two additional cases: where the resonator is pumped with 30 comb lines and 25 comb lines (the dispersion between the chips adjusted accordingly) while the same average on-chip power is maintained at 11 mW in the bus waveguide of the resonator chip. Below 25 comb lines, the required relative microwave powers between amplitude and phase modulation can no longer be achieved by our measurement scheme (the amplitude modulator is driven below the half wave voltage with the variable attenuator set to 0).

Response Figure 2. Study of frequency comb generation for different spans of pump EO comb lines with 11 mW average power on-chip. Inset: pump EO comb for each spectrum. In each case, four wave mixing still dominates over the typical Raman scattering that is seen in the CW-pumped case.

We see that even with a near 40% reduction in the number of pump comb lines, the broadening threshold in the pulse-pumped scheme is still dominant over stimulated Raman scattering. While going down in pump lines for the existing scheme would eventually reach a limit where the peak power does not reach the Raman threshold, we believe that this trend should hold down to the case where single digit comb lines are used to pump the resonator.

The reviewer also brings up an excellent point that some level of SRS may be beneficial in broadband spectrum generation. Since our initial submission, another manuscript has been published on the dynamics of Raman and Kerr effects in pulse-pumped resonators⁴ (in addition to the manuscript cited by the reviewer). The features in the simulated spectra resemble those in our Fig. 3b and do suggest that some amount of SRS may aid in broadening the spectrum further in the normally dispersive regime. However, since we are not able to reproduce the same spectral features in our response measurements (due to aging of the uncladded resonator device that cannot be fully recovered through re-annealing), we choose not to make claims about this effect in our manuscript and instead add the following to our outlook with proper citations: “Careful study of the interplay between stimulated Raman scattering and soliton formation under pulsed pumping could be explored further for even broader bandwidth combs in the normally dispersive regime.”

Why do the authors expect SRS to be in the backward direction?

We expect the stimulated Raman scattering to be in the backward direction due to previous experiments conducted on thin-film LN, namely in Ref. 21 of our manuscript⁵. In this work, the Raman behavior along both crystal axes is studied using X-cut LN racetrack resonators. Notably, when a circulator is used to monitor the spectrum in both the forward and backward direction, it finds that the stimulated Raman scattering is much stronger in a counter-propagating direction to the pump along both optical polarizations (corresponding to different crystal orientations). This behavior is consistent with what has been observed in our experiment.

We realize that the directionality of the stimulated Raman scattering might not be familiar to all readers, so have added additional text to the manuscript so that the reason for the measurement in the backward direction is clearer. The changes are shown below:

“Importantly, synchronous driving overcomes strong stimulated Raman scattering (SRS) in the material, another limitation in LN-based frequency combs and $\chi^{(3)}$ OPOs based on crystalline materials. SRS has been observed in both X and Z crystal cuts of thin-film lithium niobate and extensively studied along both crystal orientations in X-cut LN, which finds that light generated through SRS in LN scatters in the backward direction to the optical pump direction. In our pulsed scheme, we excite the resonator synchronously with an EO-generated comb, thus seeding and lowering the power threshold of the four-wave mixing process.”

- It is described that the repetition rate is fine-tuned to match the microcavity's FSR. How delicate is this tuning and what are the tolerances, also regarding thermal drifts etc.? Perhaps the resonator transmission and spectra for different repetition rates/center wavelength detuning could be shown?

The reviewer brings up an excellent point. In response, we have added a section and figure to the Supplementary Material which addresses the effect of microwave detuning on the final optical

spectrum. In our experiments, we tune in 1 MHz steps and have added a note in the manuscript to reflect this: “The center frequency of the pulse source is tuned to match the resonance frequency, and its repetition rate is fine-tuned to precisely match the FSR of the Kerr ring with 1 MHz accuracy (see Supplementary Material).”

We note significant sensitivity to the repetition rate on the optical spectrum, which the reviewer will see reflected in our figure. In fact, 2-3 MHz detuning is sufficient to see a significant effect on the frequency comb spectrum. Additionally, spectrum stability decreases significantly with increased repetition rate detuning. Please see the figure below and refer to the supplementary section titled “Repetition rate detuning.”

Response Figure 3. Effect of pulse source repetition rate detuning on the broadened spectrum. The resonator is pumped at equivalent average power and peak power with 1 MHz steps in the pulse repetition rate, from 0 MHz detuning to 3 MHz detuning.

- Is there any enhancement of the effective χ^3 nonlinearity through cascaded χ^2 effects?

While an interesting question, we have reason to believe that the contribution to the effective $\chi^{(3)}$ nonlinearity from cascaded $\chi^{(2)}$ effects is negligible and not relevant to the scope of this paper, which we will discuss below.

The reviewer will know that the effective third order nonlinearity from subsequent up- and down-conversion to 2ω and back to ω is proportional to $|\chi^{(2)}|^2/\Delta k$, where Δk is the wavevector mismatch in the second order process⁶. We note that in the case of the TE mode Z-cut resonator, the light is only polarized along the ordinary crystal axis, meaning that we are only able to access the d_{22} tensor component of the $\chi^{(2)}$ LN tensor. LN's d_{22} is an order of magnitude lower than d_{33} , which can be utilized for light polarized along the extraordinary crystal axis ($d_{22} = 2.1$ pm/V, $d_{33} = 27.0$ pm/V at 1064 nm⁷). Additionally, there has been no concerted effort by us to engineer the material cross-section to minimize Δk for strong $\chi^{(2)}$ processes.

References

1. Anderson, M. H. *et al.* Photonic chip-based resonant supercontinuum via pulse-driven Kerr microresonator solitons. *Optica* (2021).
2. Gong, Z., Liu, X., Xu, Y. & Tang, H. X. Near-octave lithium niobate soliton microcomb. *Optica* **7**, 1275 (2020).
3. Yu, M. *et al.* Integrated femtosecond pulse generator on thin-film lithium niobate. *Nature* **612**, 252–258 (2022).
4. Li, Z. *et al.* Ultrashort dissipative Raman solitons in Kerr resonators driven with phase-coherent optical pulses. *Nat. Photonics* **18**, 46–53 (2023).
5. Yu, M. *et al.* Raman lasing and soliton mode-locking in lithium niobate microresonators. *Light Sci Appl* **9**, 9 (2020).
6. Stegeman, G. I., Hagan, D. J. & Torner, L. $\chi(2)$ cascading phenomena and their applications to all-optical signal processing, mode-locking, pulse compression and solitons. *Opt. Quantum Electron.* **28**, 1691–1740 (1996).
7. Zhu, D. *et al.* Integrated photonics on thin-film lithium niobate. *Advances in Optics* (2021).

REVIEWER COMMENTS

Reviewer #1 (Remarks to the Author):

The revision addressed all pertinent points. The point with backward SRS mitigation is very important in the regime where normal dispersion cannot sustain Raman soliton.

Some remaining comments

- References seem to be incomplete.
- The phonon mode attribution needs to be checked. According to the measurements in Ref. 36, for the scattering geometry relevant to this work, the mode mixture of E LO8 and A1 TO4 is more likely. There is no Raman scattering configuration that would allow to obtain pure E(LO) modes (see, e.g., N. Kokanyan Ph.D. thesis "Study of photo-electrostrictive effects in photorefractive LiNbO3 probed by polarized Raman spectroscopy," from Université de Lorraine, 2015).
- The SRS threshold increases if the pump pulse length is shorter than the relevant phonon relaxation time (\sim inverse of the Raman bandwidth). That might be a good starting point for discussing possible SRS mitigation strategies.

Reviewer #2 (Remarks to the Author):

The authors have addressed most points and I believe the observation of suppressed Raman scattering is important; before publication, I believe further clarification is needed regarding the on-chip femtosecond pulse source. As previously mentioned, the fs pulse generator is not entirely on-chip, as it involves off-chip compression (and amplification). Although one could call the intensity-modulated signal a pulse, it does not fulfill the requirements for pumping the cavity. Compressing the modulated signal into the actual pump pulse necessitates several tens of meters of optical fiber. This critical detail is not adequately represented in the manuscript, leading to potential misunderstandings. For instance, the abstract mentions "... using a tunable on-chip femtosecond pulse generator" which is not correct. In Figure 1b, the necessary dispersion/compression post modulation is omitted, although it is an essential component. In the conclusion, the authors anticipate full integration that also includes on-chip pulse compression. An illustration of a Bragg grating in Figure 4a hints at a possible approach, but this is neither mentioned in the text nor supported by references to gauge its practicality. In my view this should still be addressed.

Response to the Reviewers

We would like to thank the reviewers for reviewing the revisions to our manuscript and providing additional feedback, based on which we have made more changes to the manuscript. We address the reviewers point by point below.

Reviewer #1 (Remarks to the Author):

The revision addressed all pertinent points. The point with backward SRS mitigation is very important in the regime where normal dispersion cannot sustain Raman soliton.

We thank the reviewer for the positive response to the changes in our manuscript.

Some remaining comments

- References seem to be incomplete.

To remedy this point, we have added several additional references to our manuscript, which address previous work on Kerr comb generation, electro-optic-based pulse generation, and Raman scattering. We have also moved some references forward in the paper which were cited later in the manuscript but got accidentally shuffled around in the revision stage. If we are missing additional references, we will be happy to add them to the manuscript at the reviewer's request.

- The phonon mode attribution needs to be checked. According to the measurements in Ref. 36, for the scattering geometry relevant to this work, the mode mixture of E LO₈ and A₁ TO₄ is more likely. There is no Raman scattering configuration that would allow to obtain pure E(LO) modes (see, e.g., N. Kokanyan Ph.D. thesis "Study of photo-electrostrictive effects in photorefractive LiNbO₃ probed by polarized Raman spectroscopy," from Université de Lorraine, 2015).

We thank the reviewer for pointing out a potential error in the manuscript. We have carefully reviewed various references to provide an accurate phonon mode attribution. The source of the error is attributed to previously published work on TE-polarized Z-cut TFLN which attributes the 625 cm⁻¹ frequency shift to the E(LO)₈ mode. Upon further examination of the literature in bulk lithium niobate, the E(TO)₉ mode is the most likely based on the calculated frequencies in bulk literature and the crystal orientation of our waveguide with respect to the light polarization. The table below summarizes the measured E(TO)₉ frequency shifts in literature, calculated and experimental, which we believe is sufficient evidence that we have correctly identified the phonon mode in our system. We have updated the text accordingly and removed the reference to the other work on Z-cut TFLN from that section of the manuscript. Additional references have also been added along with the mode attribution.

Reference	E(TO) ₉ Frequency (cm ⁻¹)
Caciuc, V., Postnikov, A. V. & Borstel, G. Ab initio structure and zone-center phonons in LiNbO ₃ . Phys. Rev. B Condens. Matter 61 , 8806–8813 (2000).	617

Sanna, S. et al. Raman scattering efficiency in LiTaO ₃ and LiNbO ₃ crystals. Phys. Rev. B Condens. Matter Mater. Phys. 91 , (2015).	661
Ridah, A., Bourson, P., Fontana, M. D. & Malovichko, G. The composition dependence of the Raman spectrum and new assignment of the phonons in. J. Phys. Condens. Matter 9 , 9687 (1997).	609.8
Repelin, Y., Husson, E., Bennani, F. & Proust, C. Raman spectroscopy of lithium niobate and lithium tantalate. Force field calculations. J. Phys. Chem. Solids 60 , 819–825 (1999).	610
Kaminow, I. P. & Johnston, W. D. Quantitative Determination of Sources of the Electro-Optic Effect in LiNbO ₃ and LiTaO ₃ . Phys. Rev. 160 , 519–522 (1967).	630

- The SRS threshold increases if the pump pulse length is shorter than the relevant phonon relaxation time (~inverse of the Raman bandwidth). That might be a good starting point for discussing possible SRS mitigation strategies.

We have used the suggestion of the reviewer in the introduction the benefits of operating in the pulsed-pumping regime for resonators, adding the additional feature of SRS mitigation: “Synchronous pumping can lower threshold power for parametric oscillation due to high pulse peak power and has been shown to enable broadband **on-chip** frequency comb generation of much lower (10s of GHz) repetition rates. **Driving with ultrashort pulses could also be an alternative means of mitigating stimulated Raman scattering in microresonators, by simultaneously increasing the SRS threshold when the pulse duration is shorter than the phonon relaxation time and decreasing the four-wave mixing (FWM) threshold by seeding the nonlinear process.**”

Reviewer #2 (Remarks to the Author):

The authors have addressed most points and I believe the observation of suppressed Raman scattering is important.

We thank the reviewer for the positive response to the changes in our manuscript.

Before publication, I believe further clarification is needed regarding the on-chip femtosecond pulse source. As previously mentioned, the fs pulse generator is not entirely on-chip, as it involves off-chip compression (and amplification). Although one could call the intensity-modulated signal a pulse, it does not fulfill the requirements for pumping the cavity. Compressing the modulated signal into the actual pump pulse necessitates several tens of meters of optical fiber. This critical detail is not adequately represented in the manuscript, leading to potential misunderstandings. For instance, the abstract mentions "... using a tunable on-chip femtosecond pulse generator" which is not correct. In Figure 1b, the necessary

dispersion/compression post modulation is omitted, although it is an essential component. In the conclusion, the authors anticipate full integration that also includes on-chip pulse compression. An illustration of a Bragg grating in Figure 4a hints at a possible approach, but this is neither mentioned in the text nor supported by references to gauge its practicality. In my view this should still be addressed.

The reviewer brings up an excellent point, and we have implemented changes to the manuscript to address this. We first make the following change to Figure 1, which now includes the dispersion component as an important component for synchronous pulsed pumping, as well as a visualization of the shorter pulse durations enabled via the dispersion, along with the modified caption. The accompanying main body text now reads, “Pulses are generated from CW light using an electro-optic time lens system through amplitude and phase modulation and dispersion, before being sent to the nonlinear resonator.”

Fig. 1 Visualization and concept of pulse and frequency comb generation. **a**, CW scheme in time and frequency domain; CW light is tuned to the resonant frequency of a microresonator and undergoes four-wave mixing process. The generated frequency comb pulse train has a repetition rate equal to the free spectral range of the microresonator; **b**, Pulse generation and synchronous pumping scheme in time and frequency domain; a pulse train is formed through amplitude and phase modulation of CW light and compressed in a dispersive medium before resonantly driving the microresonator.

Additionally, we change the text to reflect that the femtosecond pulse generator is not entirely on-chip, but rather that the pulse generation system includes on-chip modulation. The abstract is modified to read: “Here, we take steps to overcome this limitation and demonstrate broadband frequency comb generation using a $\chi^{(3)}$ pulse-driven resonator by using a tunable femtosecond pulse generator **with on-chip amplitude and phase modulators** to synchronously pump the ring.” Additionally, when introducing the work, we write instead: “In this work, we demonstrate a broad 30-GHz Kerr comb source by leveraging $\chi^{(3)}$ nonlinearity of dispersion-engineered high- Q thin-film LN microresonator that is synchronously pumped using an electro-optic femtosecond pulse source **realized via on-chip electro-optic modulation** on thin-film LN **with pulse compression in optical fiber**. The on-chip **modulation** not only reduces the footprint of our pulsed comb generation scheme, but also gives flexibility of the pulse repetition rate and center wavelength while maintaining small form factor.”

In the outlook for the manuscript, we add points to address the feasibility of integrated implementation more explicitly. In discussing the benefits of full integration, we say, “This would not only greatly reduce the footprint of the comb generator, but also allow us to avoid on- and off-chip coupling losses between the two chips. **Combined with higher pump-power lasers, this loss reduction would also eliminate the need for an intra-system amplifier.**” We also add a paragraph to explicitly discuss on-chip dispersion compensation:

An outstanding bottleneck to a fully integrated scheme not demonstrated in this work is the need for on-chip dispersion compensation to compress the pulses generated by the modulator system. One approach to on-chip dispersion is through the use of a long waveguide with large engineered dispersion. However, because of the waveguide geometries usually needed to achieve high dispersion, this approach is limited largely by the propagation loss in these waveguides, currently much higher than the state-of-the-art TFLN propagation loss. Waveguide gratings are a more commonly used method for dispersion compensation on integrated photonics platforms and can achieve large dispersion factors over a small propagation distance; they have been implemented on the TFLN platform with low insertion loss and small footprint. Because these gratings operate in the reflection mode, most implementations of on-chip dispersion gratings require optical circulators or beam splitters. Recently, circulator-free dispersion schemes based on coupled waveguide gratings (used to illustrate a potential approach for on-chip dispersion in Fig. 4a) and mode de-multiplexers have been proposed and demonstrated.